# Mycn Is Essential for Pubertal Mammary Gland Development and Promotes the Activation of Bcl11b-Maintained Quiescent Stem Cells

**DOI:** 10.3390/cells14161239

**Published:** 2025-08-12

**Authors:** Zuobao Lin, Chunhui Wang, Huiru Bai, Yue Zhang, Meizhen Lin, Xiaoqin Liu, Tian’en Hu, Yuan Meng

**Affiliations:** 1Westlake Laboratory of Life Sciences and Biomedicine, School of Life Sciences, Westlake University, Hangzhou 310024, China; wangchunhui@westlake.edu.cn (C.W.); baihuiru@westlake.edu.cn (H.B.); zhang_yue@westlake.edu.cn (Y.Z.); linmeizhen@westlake.edu.cn (M.L.); liuxiaoqin@westlake.edu.cn (X.L.); hutianen@westlake.edu.cn (T.H.); mengyuan@westlake.edu.cn (Y.M.); 2Westlake Institute for Advanced Study, Fudan University, Hangzhou 310024, China; 3Westlake Disease Modeling Lab, Westlake Laboratory of Life Sciences and Biomedicine, Hangzhou 310024, China; 4Key Laboratory of Growth Regulation and Translational Research of Zhejiang Province, School of Life Sciences, Westlake University, Hangzhou 310024, China

**Keywords:** Mycn, activation, quiescent, mammary stem cells, Bcl11b, Tspan8

## Abstract

This investigation examines the function of the mouse *Mycn* gene in regulating and activating quiescent mammary stem cells, which are vital for mammary gland development. The mammary gland, consisting of luminal and basal cells, progresses through complex developmental stages from embryonic development through puberty, adulthood, pregnancy, lactation, and involution. Quiescent stem cells, existing in a reversible non-proliferative state, are essential for gland maintenance, yet their activation mechanisms remain poorly understood. Mycn, a member of the Myc/MYC oncogene family, is recognized for its roles in embryonic development and cancer, notably aggressive neuroblastoma and triple-negative breast cancer. Through single-cell RNA sequencing (scRNA-seq), CRISPR knockout, and overexpression experiments, this study demonstrates that Mycn is highly enriched in the terminal end buds (TEBs) of the pubertal mammary gland, particularly in basal cells, and is critical for ductal development. Both deletion and overexpression of *Mycn* diminish the stemness and regenerative capacity of mammary stem cells. Mycn enhances cell proliferation while downregulating quiescent stem cell markers and regulators, including Bcl11b and Tspan8, affecting stem cell maintenance and differentiation. This research clarifies the regulatory role of Bcl11b in controlling Tspan8 expression and demonstrates that Mycn indirectly targets both under normal conditions. Maintaining appropriate levels of Mycn expression is essential for normal development and cancer prevention. These insights contribute to understanding diseases and aggressive cancers, including triple-negative breast cancer (TNBC), and suggest potential therapeutic approaches.

## 1. Introduction

Cancer treatment has evolved significantly through three main periods [1]: The first period is pre-1970, when traditional modalities established the foundation of cancer therapy. This included surgery, radiation therapy, chemotherapy, and bone marrow transplantation, which formed the cornerstone of oncological interventions for decades. The second period spans from 1970 to 2023, during which molecular understanding revolutionized treatment approach development. This era saw the clinical use of Tamoxifen, monoclonal antibodies, immune checkpoint inhibitors, and CAR-T therapy, all representing significant advances based on deeper comprehension of cancer biology. Future directions include new approaches, such as advanced immunotherapies and personalized cancer vaccines. CRISPR-mediated gene therapy, as well as AI-assisted diagnosis and treatment planning, also show great promise for improving cancer management in coming years [1].

Despite these significant advances in cancer treatment, certain types, including triple-negative breast cancer, remain difficult to cure, primarily due to the lack of targetable receptors and high molecular heterogeneity, which limit the effectiveness of current precision medicine approaches [2,3,4,5].

Despite significant advances in triple-negative breast cancer research in recent years, we have yet to fully overcome this challenging disease. At its core, this limitation reflects our insufficient understanding of how the regulatory networks that govern normal breast tissue developmental become dysregulated, as well as the molecular mechanisms underlying tumor initiation and progression. Elucidating the regulatory functions of key transcription factors such as Mycn in these processes may provide a breakthrough for developing novel therapeutic strategies [6,7,8,9,10].

In this study, we employ the mouse as an experimental model. The mammary gland is a specialized organ in mammals for milk production and neonatal nourishment [11]. The development of the mammary gland in mice encompasses a complex process that begins during the embryonic stages and continues through puberty, adulthood, pregnancy, lactation, and involution, cycling repeatedly [11].

The mammary gland primarily comprises two main cell types: an inner layer of luminal cells and an outer layer of elongated, contractile basal cells [11,12,13]. Basal cells are myoepithelial cells characterized by the expression of basal keratin 5 (K5) and keratin 14 (K14), possessing contractile properties and functioning to define the basal membrane and encapsulate the ductal-facing luminal cells. Luminal cells, on the other hand, express keratin 8 (K8) and keratin 18 (K18) [11,14]. These cells exhibit polarity and can be further clarified into estrogen receptor (ESR1 or ER)-positive and -negative ductal cells and alveolar cells, which facilitate milk production [11,15,16].

During embryonic development, mammary epithelial cells consist of a distinct population of undifferentiated and highly plastic progenitor cells that ultimately generate all postnatal mammary epithelial cells [17,18,19]. The postnatal mammary gland maintains the ability for rapid ductal expansion during puberty. At puberty onset, bulb-shaped structures called the terminal end buds (TEBs) form at the tips of growing epithelial ducts and drive ductal elongation, invasion, and branching of the mammary trees [11,20,21,22]. TEBs comprise two cell types: cap cells, which serve as progenitors of myoepithelial cells, and body cells, which function as progenitors of luminal cells in the subtending ducts [11,23,24,25,26,27,28]. When TEBs reach the fat pad edges, they regress. Upon completion of puberty, TEBs dissolve, establishing the mature adult mammary gland. This gland comprises an epithelial framework, which provides the foundation for the development of alveoli-like bud development during the estrus cycle and alveoli formation during pregnancy [11,28,29].

The dynamic changes in mammary epithelial cell proliferation, differentiation, and turnover during the process of mammary gland development and multiple pregnancies suggest the existence of mammary stem cells [11]. The mammary fat pad transplantation assay represents a gold-standard technique for studying mammary epithelial organogenesis and is visualized through carmine-alum whole-mount staining or bioluminescence imaging [30,31]. The implementation of classical mammary stem cell transplantation assays and lineage tracing analysis in model animals has demonstrated the presence of undifferentiated stem cells in the postnatal mammary organ [32,33].

In recent years, researchers have identified several markers to characterize mammary stem cells, including s-SHIP^+^ [28], Axin2^+^ [34], Cd1d^+^ [35], Procr^+^ [36], Bcl11b^high^ [12], Lgr5^+^Tspan8^high^ [13], and Dll1^high^ [37]. These markers identify distinct types of stem cells that serve crucial roles in mammary gland development, homeostasis, and regeneration. The presence of these diverse stem cells demonstrates the heterogeneity within stem cell populations in the mammary gland system. This research primarily focuses on quiescent mammary stem cells and their biological functions.

Quiescence refers to a reversible state of the cell cycle, typically occurring in the G0 phase, though it occasionally occurs in the G2 phase [38,39]. While non-proliferation characterizes both terminally differentiated and senescent cells, only quiescent cells retain the capacity to rejoin the cell cycle under appropriate physiological conditions [40]. This quiescent state proves crucial for the longevity of proliferative cells, as it helps prevent the depletion of their proliferative capacity and reduces the risk of accumulating damage to DNA, proteins, and mitochondria, which could lead to malignant transformation or senescence [40].

Dysregulated exit from quiescence has been linked to hyperplasia and tumorigenesis, highlighting the importance of proper quiescence regulation [41,42,43]. The transcription factor Bcl11b has emerged as a critical regulator of this process in mammary tissue, functioning as a molecular switch between normal cellular aging and tumorigenesis [41]. Studies have demonstrated that Bcl11b deficiency accelerates the aging phenotype, characterized by upregulation of age-related signaling pathways (including MAPK, PI3K-Akt, NF-κB, and p53), enhanced cellular senescence, and activation of the senescence-associated secretory phenotype (SASP). Notably, Bcl11b loss leads to distinct early and late senescence states (e-Sen and l-Sen), with l-Sen cells exhibiting heightened NF-κB activity and increased susceptibility to malignant transformation. Mechanistically, Bcl11b directly suppresses NF-κB signaling and other aging-associated pathways, serving as a protective factor against premature senescence. The therapeutic relevance of this regulatory axis is underscored by experiments showing that inhibition of NF-κB/Jak-Stat signaling with TPCA-1 can rescue the aging phenotype and significantly reduce tumor incidence in Bcl11b-deficient models. These findings establish Bcl11b as a crucial determinant of mammary cell fate, governing the balance between quiescence, senescence, and neoplastic transformation, and suggest that targeting senescence programs may offer preventive strategies for age-related breast cancer [41].

Recent research has identified Bcl11b^high^ [12], Lgr5^+^Tspan8^high^ [13], and Nfatc1^+^ [44] basal cells as quiescent stem cells. Studies indicate that the Bcl11b^high^ quiescent mammary epithelial cell population expresses elevated levels of Bcl11b and resides at the interface between luminal and basal cells, while the Lgr5^+^Tspan8^high^ mammary stem cells represents a deeply quiescent subset that resides within the proximal region [13]. The dormant Nfatc1^+^ basal stem cells exhibit distinct characteristics from other known mammary stem cells [44]. Nevertheless, they all serve essential roles in long-term maintenance of the mammary gland.

The molecular mechanisms governing mammary stem cell activation from quiescence remain incompletely understood. Previous research identified Foxp1 as a direct repressor of Tspan8 in basal cells, potentially regulating mammary stem cell dormancy exit during differentiation and development [45]. In contrast to Bcl11b [12] or Foxp1 [45], Zeb1 distinctively promotes self-renewal in a specific subpopulation of basal stem cells while inhibiting their proliferation, as documented by the author [46]. Given the heterogeneous nature of mammary stem cells, the precise mechanism activating Bcl11b^high^ quiescent stem cells remains unclear. Our recent research has identified Mycn, a cell proliferation regulator, as a crucial factor controlling the exit from mammary stem cell quiescence.

The discovery and understanding of the mouse *Mycn* gene in mice originated from the study of the human *MYCN* gene. *MYCN* belongs to the *MYC* oncogene family, which consists of three members (*MYC*, *MYCN*, and *MYCL*) and was first identified in neuroblastoma [47,48,49].

The *MYC* family serves a fundamental role in cellular processes and frequently contributes to cancer development. MYC proteins function as primary drivers of tumorigenesis, with dysregulated levels observed in numerous human tumors [50,51,52,53,54,55,56,57,58,59,60,61,62,63]. MYCN, encoded by the *MYCN* gene, plays an essential role in embryonic development, particularly in regulating growth, proliferation, and metabolism of progenitor cells across various developing organs and tissues [48,49,64]. Research demonstrates that the protein encoded by *MYCN* binds to specific promoters, influencing the expression of crucial target genes involved in multiple cellular functions, consistent with studies in mammalian cells [65]. Its dysregulation, particularly overexpression, correlates with various cancers, notably neuroblastoma, a pediatric cancer originating from immature nerve cells [66]. These findings underscore the importance of *MYCN* in regulating cellular processes that can become dysregulated in cancer.

The *MYC* family genes demonstrate crucial functions in stem cell biology. *N-MYC* (*MYCN*) and C-*MYC* have been identified as essential regulators of both neural and hematopoietic stem cell quiescence and activation, including proliferation, differentiation, and long-term self-renewal activity [67,68,69,70,71,72,73,74,75,76]. Despite the complex nature of *MYC* family regulation and function, their significance in cancer and cell biology has established them as central subjects in biological and oncological research. Ongoing efforts to understand and target the *Myc* family, particularly *N-myc* (*Mycn* in mouse), continue to enhance our understanding of cancer biology and facilitate novel therapeutic approaches.

Research indicates that the MYC family, including MYCN, correlates with breast cancer tumorigenesis [55,77,78,79,80,81], particularly in triple-negative breast cancer (TNBC) [82,83,84,85,86]. While the mechanisms of MYC in promoting breast cancer, including TNBC, have been extensively investigated, its role in normal mammary gland development requires further elucidation. Based on our research interests and preliminary findings, this study focuses primarily on examining mouse *Mycn*’s role in maintaining mammary gland homeostasis through the regulation of quiescent mammary stem cell activation.

## 2. Materials and Methods

### 2.1. Mice

The *Bcl11b^flox/flox^* mice (C57BL/6 background) were generously provided by Mark Leid’s lab and were described previously [87]. The *Krt14-Cre* (stock number 018964), *mTmG* (stock number 007676) mice were purchased from Jackson Laboratory. For analysis of pubertal and adult mammary glands, 5~6-week-old and 2~6-month-old female mice were used, respectively. For each experiment, the ages of the female mice used were indicated in the text. In addition, for the *Bcl11b* KO assay, the mice had a genotype of *Krt14-Cre* (*+*) *mTmG* (*+*, *+*) with or without a homogeneous *Bcl11b* (*loxp*, *loxp*) background for Bcl11b deletion. All mice were housed in specific pathogen-free conditions and bred at the Westlake University Laboratory Animal Resources Center. All animal experiments were approved by the local institutional animal ethics board (permission number: 24-149-CS-3).

### 2.2. Tissue Processing and Flow Cytometry

Mammary glands from pubertal (5–6 weeks) and adult (>2 months) virgin or pregnant C57BL/6 mice were dissected and processed according to the published protocol [12].

### 2.3. Colony Formation Assay

To perform the colony formation assay, 30 μL of growth factor reduced Matrigel (BD Bioscience, San Jose, CA, USA) was added into the 96-well round-bottom plate and then solidified at 37 °C for 10 min. A total of 4K/well cells were resuspended in 200 μL culture media (DMEM/F12 + 2% FBS+ 1%P/S + B27 + 10 mM HEPES) supplemented with EGF (10 ng/mL, BD Bioscience, San Jose, CA, USA), Rspo1 (250 ng/mL, R&D, Hong Kong SAR, China), and ROCK inhibitor Y27632 (10 μm, Sigma, Hong Kong SAR, China) and were overlaid on top of the matrigel. The plate was maintained in a 37 °C incubator with 5% CO_2_ for 1–2 weeks.

### 2.4. Transplantation Assay

Transplantation assays were performed as previously described [12]. The cells were sorted and then resuspended in injection media (DMEM/F12 + 50% Matrigel + 1% P/S) at a concentration of 50,000 cells/5 μL, followed by serial dilutions to 10,000 cells/5 μL, 1000 cells/5 μL, 200 cells/5 μL, and 50 cells/5 μL. The cells were then injected into the cleared fat pads of 3-week-old recipient mice (C57BL/6) at the specified dilutions. After six to eight weeks post-transplantation, the recipient mice were analyzed through whole-mount staining of the mammary glands with carmine. The frequency and confidence interval of MRUs (mammary repopulating units) were calculated using ELDA (http://bioinf.wehi.edu.au/software/elda/ (accessed on 15 October 2024)).

### 2.5. Real-Time PCR

Between 50 and 500 primary mammary cells or cultured epithelial cells were directly sorted into 400 μL Trizol (Life Technologies, Van Allen Way, Carlsbad, CA, USA). The detailed method was conducted as previously described [12]. Data analysis was performed using Graphad 8 and Excel with normalization to the expression of *β-actin*. The value of one reference sample was set to 1 to enable inter-sample comparisons. Statistical analysis employed two-tailed unpaired *t*-test. Data are presented as mean ± SEM. The SYBR Green primers are listed below:β-Actin: Forward ACCTTCTACAATGAGCTGCG, Reverse CTGGATGGCTACGTACATGG;Bcl11b: Forward AGGAGAGTATCTGAGCCAGTG, Reverse GTTGTGCAAATGTAGCTGGAAG;Tspan8: Forward AGTTCCGTTTACCCAAAGACC, Reverse GCACCATAGAAAACACCAAACC;Mycn: Forward GTCTGTTCCAGCTACTGCC, Reverse TCCTCTTCATCTTCCTCCTCG.

### 2.6. Paraffin Sections and Immunofluorescence

The method has been previously described [12].

### 2.7. Carmine Stain (Whole Mount)

The method has been previously described [12].

### 2.8. Mycn-Overexpressing Cell Line

Primary mammary basal cells from 5–6-week-old C57BL/6 mice were collected in Trizol (Life Technologies, Van Allen Way, Carlsbad, CA, USA), and RNA was extracted following the standard protocol for reverse transcription to obtain cDNA using Superscript III First Strand Synthesis kit (Life Technologies, Van Allen Way, Carlsbad, CA, USA) according to the manufacturer’s instructions. The *Mycn* (gene ID:18109) coding sequence was amplified from cDNA and subsequently cloned into pCDH plasmid (72484) as an overexpression construct. Lentivirus production involved co-transfecting 293T cells with the overexpression construct, pCMV-dR8.2 plasmid, and pCMV-VSVG plasmid at a ratio of 10:9:1. Viral supernatants were collected at 48 h and 72 h, then concentrated through ultracentrifugation at 20,000 rpm for 2 h at 4 °C. Viral titration was performed by infecting 293T cells with gradient virus for titer calculation. Successfully infected cells were isolated by FACS based on GFP expression.

### 2.9. CRISPR-Mediated Mycn Knockout Primary Basal Cells

To verify the function of the candidate gene, knockout cell lines were generated using the CRISPR-Cas9 system. gRNAs for *Mycn* knockout were designed using an online CRISPR Design Tool http://chopchop.cbu.uib.no/ (accessed on 10 October 2023). gRNA sequences were cloned into lentiCRISPRV2 plasmid (Addgene#52961) by BsmBI. The lentiCRISPRV2 plasmid containing Cas9 sequence and gRNA scaffold was packaged into lentivirus for primary mammary basal cells infection as described previously. The cells were selected using Puromycin (2 μg/mL) following lentivirus infection. Selected mammary basal cells were cultured on the cell culture plates coated with growth factor-reduced Matrigel Matrix (Corning) for colony formation. Individual colonies were selected under microscope and cultured for expansion. Indels were analyzed using the online tool TIDE (https://tide.nki.nl/, accessed on 10 October 2023, using R version 3.3.2.). After individual colonies were picked, they were expanded through further culture and subjected to sequencing, as well as Western blotting, to confirm successful *Mycn* knockout.

### 2.10. Single-Cell RNA Sequencing (scRNA-seq) and Data Analysis

Single cells dissociated from the nipples, ducts, and TEBs of mouse mammary glands were isolated through FACS (MoFlo Astrios, Beckman Coulter, Hong Kong SAR, China) into individual wells of 96-well PCR plates containing preloaded lysis buffer, an ERCC spike-in, and a barcode. Libraries were constructed according to previously established protocols [88]. The single-cell data were deposited in GEO: GSE251933.

An analysis of reported scRNA-seq data from CD49f^high^ EpCAM^low^ Lin^−^ cells in WT-Basal and *Bcl11b*-KO C57BL/6 mice [41] revealed Bcl11b positively regulated genes (genes with *p* value < 0.05 & KO < WT intersecting with Bcl11b targets) and negatively regulated genes (genes with *p* value < 0.05 & KO > WT intersecting with Bcl11b targets), including Tspan8.

### 2.11. D-CUT&RUN

The ChIP-seq assay was performed to identify Mycn target genes, as previously described [89].

### 2.12. Data Analysis and Related Software

Stochastic analysis was conducted using Graphad Prism 9. Comparisons between two groups utilized were used for statistical analysis. Data were compared between two groups of samples using the unpaired, two-tailed Student’s *t* test. Sample sizes for each experiment were determined based on a minimum of *n* = 3 independent devices per experimental group. All data are presented as mean ± SEM. * *p* < 0.05, ** *p* < 0.01 and *** *p* < 0.001. Primer design was accomplished using Snapgene 7, and FACS data analysis was performed with Flowjo 10. Picture analysis utilized Image J 1.55 and Photoshop 2021. Adobe Illustrator CS6 was employed for image layout and editing.

## 3. Results

### 3.1. Mycn Was Spatially Enriched in TEB

Single-cell RNA sequencing (scRNA-seq) analysis was performed to examine the expression of *Mycn* in the TEB, duct, and nipple of the pubertal mammary gland (Figure 1A). The scRNA-seq data demonstrated significant *Mycn* enrichment in TEB basal cells, with lower levels in duct basal cells (Figure 1B), while *Mycn* expression in nipple basal cells was virtually undetectable (Figure 1B). Additionally, the expression of *Ki67(Mki67)* in scRNA-seq data revealed that the basal cells in the nipple exhibited relative quiescence, contrasting with the basal cells in the duct and TEB (Figure 1C). Furthermore, TEB basal cells displayed the highest proliferation characteristics (Figure 1C). These findings align with previous research, indicating that quiescent mammary stem cells primarily reside in the nipple region [13], and that TEB represents a highly proliferative region of the pubertal mammary gland [11,21]. High expression of *Mycn* in TEB basal cells was subsequently validated at both transcriptional and protein levels (Figure 1D–F). ScRNA-Seq data from Tabula muris indicated that Mycn-positive cells were present in the basal cells but absent in the luminal cells of the adult mouse mammary gland (Appendix A).

Subsequently, we verified the expression of *Mycn* in adult mammary ducts (Appendix A, the upper panel) and early pregnant alveolar structures (Appendix A, the lower panel). These observations suggest the persistent presence of Mycn-positive cells and their potential role in maintaining adult mammary homeostasis and reproductive cycles, warranting further investigation. A key question emerges regarding whether Mycn-positive cells in TEBs during puberty are associated with basal cells in alveolar structures during pregnancy. Notably, Lixia Bai and Larry R. Rohrschneider studied *s-SHIP^high^* cap cells enriched in TEB and discovered that these cells exhibited characteristics of activated stem cells with self-renewal and regenerative capabilities, proliferating rapidly in pubertal TEBs and pregnant alveolar structures [18]. This finding suggests a potential role for Mycn-positive basal cells in mouse reproductive cycles.

Unlike *Mycn*, which is expressed exclusively in basal cells, *Foxp1* shows high expression in most basal cells and many luminal cells (Appendix A). This broad expression complicates isolation of the specific effects of Foxp1 loss in mammary stem cells on gland expansion and luminal lineage differentiation.

### 3.2. Mycn Was Indispensable for Mammary Ductal Development

The pioneering studies by DeOme and his students, Les Faulkin and Charles Daniel, utilized serial transplantation of normal mammary glands into the cleared mammary fat pads of syngeneic mice (also called limiting dilution transplantation assay) to assess the ratio of mammary stem cells [90,91]. This technique allowed normal mammary cells to grow in their natural anatomical location and under a relatively normal physiological environment. Their experiments demonstrated that the mammary gland contained stem cells capable of developing into a fully functional ductal mammary tree and responding to hormonal signals through normal differentiation.

Given that *MYCN* plays significant roles in various cancer types [47,48,49,50], including triple-negative breast cancer [86], we sought to determine its importance in normal mammary gland development. To investigate the function of *Mycn* in murine mammary gland development, we employed CRISPR knockout (Figure 2A) and overexpression (Figure 3A) techniques in combination with a mammary cell transplantation assay.

To evaluate our hypothesis regarding the impact of *Mycn* knockout (KO) on mammary gland function, we examined the mammary tree reconstitution capacity of *Mycn*-mutant basal cells through a limiting dilution transplantation assay. The results demonstrated that *Mycn* deletion significantly reduced the frequency of mammary repopulating units (MRUs) in the mutant group (1/3639, ranging from 1/1158 to 1/11,433) compared to the control group (1/309, ranging from 1/144 to 1/662) (Figure 2B–D), with both groups labeled by GFP. Moreover, mammary trees reconstituted by *Mycn* KO basal cells exhibited significantly reduced size and fewer branches compared to the control group (Figure 2B,C). These observations suggest that Mycn plays a vital role in mammary gland development and mammary stem cell maintenance.

Given that *Mycn* is known to be overexpressed in various types of breast cancers, including certain triple-negative breast cancers [86], we sought to examine whether *Mycn* overexpression induces abnormal mammary development. The transplantation assay of *Mycn*-overexpressing basal cells produced results similar to those of *Mycn* KO cells, demonstrating reduced mammary regeneration capacity (Figure 3B–D). Specifically, *Mycn* overexpression decreased the MRU frequency in the control group from 1/380 (ranging from 1/219 to 1/658) to 1/972 (ranging from 1/574 to 1/1648). Whole-mount images further confirmed these findings (Figure 3B,C).

These findings indicate that both *Mycn* knockout and overexpression adversely affected the stemness of mammary stem cells, resulting in diminished mammary regeneration capacity. This effect may be attributed to the dose-dependent nature of *Mycn* activity. Therefore, we proceeded to investigate the molecular mechanisms through which *Mycn*-regulated stem cell function.

### 3.3. Mycn Promoted Cell Proliferation

Previous research has demonstrated that the human gene MYCN facilitates S-phase progression and cell proliferation, particularly in tumor cells. Specifically, MYCN functions as an RNA-binding accessory factor of the nuclear exosome-targeting complex, serving a critical role in these processes [92,93]. In neuroblastoma, MYCN-induced E2F5 enhances cell proliferation by regulating cell cycle progression [94]. Moreover, MYCN and MYC directly influence tumor proliferation and tumorigenesis through BMI1 in human neuroblastomas [93,94,95,96].

To examine the role of *Mycn* in regulating mammary basal cell proliferation, we conducted colony-formation assays using *Mycn* KO and *Mycn*-overexpressing basal cells. The results demonstrated that the *Mycn* knockout resulted in reduced proliferation of basal cells in vitro compared to the control group, as indicated by the decreased colony numbers (Figure 4A,B). This difference emerged within three days, supporting the hypothesis that *Mycn* plays a vital role in mammary basal cell proliferation. Additional validation through colony formation assays with *Mycn*-overexpressing basal cells revealed increased colony formation compared to control cells, at both Day 4 and Day 6 (Figure 4C–E). The concordance between these assays emphasizes the significance of *Mycn* in mammary basal cell proliferation.

Given *Mycn*’s essential role in cell proliferation, the impairment of mammary reconstitution capacity in *Mycn* KO basal cells during transplantation was anticipated. However, the observation that *Mycn* overexpression also negatively affects mammary regeneration presents an intriguing paradox.

### 3.4. Mycn Downregulated the Quiescent Stem Cell Regulators Bcl11b and Tspan8

To elucidate the mechanism behind the decreased mammary regeneration capacity following Mycn overexpression, despite its proliferation-promoting role, quantitative PCR analysis was conducted to examine alterations in specific characteristic mammary stem/progenitor cell markers and regulators, including *Bcl11b* [12,97], *Tspan8* [13,45], and *Trp63* [18,98].

*Bcl11b*, as a transcription factor [99,100], has been identified as a crucial regulator of quiescent mammary stem cells, essential for maintaining long-term stemness, function, and differentiation [12]. Therefore, we investigated whether *Mycn* overexpression affected its expression. Additionally, we evaluated the expression level of *Tspan8*, another marker of quiescent mammary stem cells [13]. Previous research indicates that Tspan8 functions in maintaining mammary stem cell quiescence, and its deletion of Tspan8 rescued the defects in ductal morphogenesis caused by Foxp1 loss [45]. Furthermore, *Trp63* promotes unipotent basal cell fate in the embryonic mammary gland [18], while its isoform, *DeltaNp63*, enhances stem cell activity in mammary gland development through decreased *Fzd7* expression and Wnt signalling [98].

Upon examining these three markers under conditions of *Mycn* overexpression using FACS and qRT-PCR, we observed significant downregulation (Figure 5A–C). Given that *Bcl11b* KO activates mammary stem cells [12], and at the same time, deletion of *Tspan8* also facilitated the activation of quiescent mammary stem cells [45], these findings suggest that *Mycn* overexpression promotes the exit from quiescent state in mammary stem cells. Moreover, the downregulation of these key markers and regulators due to *Mycn* overexpression could potentially impair both the maintenance of stem cell properties and differentiation processes. This observation may explain the reduced outgrowth of transplanted *Mycn*-overexpressing pubertal mammary basal cells, despite *Mycn*’s role in promoting cell proliferation in vitro.

### 3.5. Bcl11b KO Decreased the Expression Level of Tspan8 and Activated Cell Proliferation

Following the observation that *Mycn* overexpression significantly reduced the expression of the quiescent stem cell regulators *Bcl11b* and *Tspan8*, we investigated whether the *Bcl11b* knockout (KO) also affected *Tspan8* expression level. We utilized the *K14-Cre/Bcl11b^flox/flox^/mTmG* mouse model for our research. Through crossing *Bcl11b^flox/flox^/mTmG* mice with *K14-Cre* mice, we generated *K14-Cre/Bcl11b^flox/+^/mTmG* mice (Figure 6A), which were subsequently backcrossed to *Bcl11b^flox/flox^/mTmG* mice. After deleting *Bcl11b* in basal cells in *K14-Cre/Bcl11b^flox/flox^/mTmG* mice, we examined the developmental effects on mammary gland development. The mammary glands of the 6-week-old Bcl11b KO mice exhibited reduced branching (Figure 6b3–b3″,b4–b4″) compared to the controls (Figure 6b1–b1″,b2–b2″). The branching data are quantified in Figure 6C. *Bcl11b* KO also resulted in reduced luminal lineage within both duct and nipple areas (Figure 6D), consistent with the observation that “The Krt14-positive cells had impaired ability to give rise to luminal cells after *Bcl11b* deletion, reminiscent of a defect in lineage commitment and multipotency at some time during the mammary gland development” [12].

To evaluate the impact of *Bcl11b* KO on *Tspan8* expression, we conducted FACS analysis to quantify the percentage of Tspan8-positive basal cells among GFP-positive basal cells, revealing a significant reduction in the nipple region and a moderate decrease in the ductal region following *Bcl11b* KO (Figure 6E,G). This pattern continued into adulthood (Figure 6F,H). Further analysis of *Tspan8* transcriptional levels using qRT-PCR in adult mammary tissues confirmed a persistent change in expression (Figure 6I). Additionally, *Bcl11b* mRNA levels were analyzed to verify that the phenotypic changes resulted from Bcl11b KO, demonstrating a similar expression pattern to *Tspan8* in both ductal and nipple basal cells of adult mammary glands (Figure 6J). An analysis of scRNA-seq data from adult (4-month-old) mice mammary glands indicated that *Bcl11b* KO in basal cells decreased *Tspan8* expression (Appendix A), a trend consistent with findings in pubertal mammary tissues. Moreover, we also detected that *Bcl11b* KO prompted an increase in basal cell proliferation (Appendix A), which can be attributed to the disruption of stem cell quiescence and the activation of basal cell proliferation.

Collectively, these findings suggested that the transcription factor *Bcl11b* played a crucial role in maintaining the balance between differentiation and stem cell quiescence during mammary gland development, with its deletion disrupting this balance, leading to impaired branching and reduced *Tspan8* level and activation of cell proliferation.

### 3.6. Mycn Coordinated Cell Proliferation and Differentiation

Given that *Bcl11b* regulated the balance between cell proliferation and differentiation and considering that *Mycn* has been demonstrated to suppress *Bcl11b* expression, we hypothesized that *Mycn* might regulate cell proliferation and differentiation processes during mammary gland development.

To test this hypothesis, we overexpressed *Mycn* gene in the mammary epithelial progenitor cell line Comma Dβ and performed a ChIP-Seq assay [89]. The analysis revealed Mycn binding sites across the genome, with 63.97% located in promoter regions in Comma Dβ cells (with 10^5^ cells) [89]. A GO analysis determined that Mycn participated in the regulation of cell cycle, growth, proliferation, and differentiation (Figure 7A–C).

These findings align with both our previous data (Figure 1C and Figure 4A–E) and established research [89,94,95,96,101,102,103,104,105,106]. Notably, we observed that Mycn directly participates in the metaphase and anaphase stages of mitosis during primary mammary basal cell division when cultured in vitro (Appendix A). These observations demonstrate that Mycn coordinates cell proliferation and differentiation.

### 3.7. Mycn Indirectly Targeted Bcl11b and Tspan8

Previous research has established Tspan8’s critical role in cancer stem cell characteristic maintenance and the maintenance of stem cell characteristics through Sonic Hedgehog signaling pathway activation [107,108]. Research has demonstrated that the human TSPAN8 protein regulates MYC expression by binding to its promoter, enhancing STAT3’s chromatin occupancy, and influencing cancer-related gene transcription, including MYC [109]. These findings indicate a potential Mycn–Tspan8 interaction. Given Tspan8’s direct interaction with Myc, it appears plausible that Mycn might similarly bind and interact with Tspan8. Additionally, as Mycn functions as a transcription factor, it may directly interact with Bcl11b to regulate cell cycle, cell proliferation and differentiation.

Research has shown that both Bcl11b and Tspan8 regulate mammary stem cell quiescence maintenance, with their inhibition or deletion promoting mammary stem/progenitor cell activation or proliferation. Having observed Mycn’s downregulation of functional stem cell markers Bcl11b and Tspan8, we investigated whether this regulation occurred directly or indirectly. We employed the improved D-CUT&RUN technology to detect Mycn binding sites in mammary epithelial progenitor cells [89]. While direct interactions between Mycn and Bcl11b or Tspan8 were not detected, the analysis of existing research revealed Mycn’s interaction with both Bcl11b and Tspan8 in mouse and human neuroblastoma tumors through promoter binding (Appendix A). This may indicate the existence of different mechanisms of interaction between Mycn and Bcl11b and Tspan8 across different organs or between normal and pathological states.

## 4. Discussion

Mammary gland development is a highly regulated, dynamic process that occurs in distinct stages, including embryonic formation, postnatal growth, puberty, pregnancy, lactation, and involution. Puberty triggers the formation of terminal end buds (TEBs) at the tips of growing ducts, driving ductal elongation and branching. These TEBs eventually regress, forming a mature mammary gland. The mature gland’s epithelial framework supports alveolar development during pregnancy and lactation. Mammary stem cells, characterized by markers such as Bcl11b^high^, Lgr5^+^Tspan8^high^, and Nfatc1^+^, play key roles in gland maintenance and regeneration, though the mechanisms for activating quiescent stem cells remain unclear.

We employ scRNA-Seq technology to help discover the heterogeneity of different mammary subpopulations, including stem and progenitor cells. We acknowledge that there are both advantages and disadvantages for scRNA-Seq and bulk sequencing. Given that bulk sequencing provides cost-effectiveness and robustness for detecting population-level expression changes but masks cellular heterogeneity, while scRNA-Seq reveals cellular diversity and rare populations but faces challenges, including increased technical noise, higher costs, and analytical biases including dropout events [110,111], a combinatory strategy would be more effective in leveraging the strengths of both approaches: using bulk sequencing for initial subtype discovery across larger cohorts, followed by strategic single-cell validation on representative samples. This approach may ensure our genomic correlations authentically reflect underlying cellular architecture while maintaining resource efficiency [112].

In our study, the transcription factor Mycn played a critical role in the exit from quiescence in mammary stem cells (Figure 8). Single-cell RNA sequencing (scRNA-seq) identified Mycn expression in TEB basal cells, validated through qPCR and immunofluorescence staining. This indicates Mycn’s significance in cell proliferation and differentiation within the TEB region. Colony formation and transplantation assays utilizing *Mycn* knockout (KO) and overexpressing basal epithelial cells established that Mycn promoted proliferation and differentiation by maintaining optimal expression levels, essential for mammary gland regeneration (Figure 8).

The data further indicate that Mycn’s interactions with Bcl11b and Tspan8 may differ between cancer and normal tissues, exhibiting a more direct interaction, observed in cancer tissues. This underscores the significance of Bcl11b and Tspan8 as downstream targets of Mycn. Notably, Bcl11b has been documented to function as a tumor suppressor in various cancers [41,113,114], aligning with the observed decrease in Bcl11b levels following *Mycn* overexpression in epithelial cells. However, TSPAN8 (human) has been shown to enhance breast cancer cell stemness through sonic hedgehog signaling activation, introducing additional complexity to understanding Mycn’s contribution to tumor progression. The potential roles and interactions of Bcl11b and Tspan8 with Mycn in cancers characterized by Mycn overexpression warrant further investigation.

Although this study primarily examined Mycn’s role in normal mammary gland development, additional research is necessary to investigate its role in differentiation using genetic mutation mouse models, as well as in vivo lineage tracing, given potential off-target effects of CRISPR. Future studies should also examine Mycn’s roles in cancer progression in both murine and human contexts.

In summary, Mycn serves essential functions in mammary gland development and regeneration by activating quiescent stem cells and promoting both proliferation and differentiation. These findings provide significant insights into cancers characterized by *Mycn* overexpression, such as triple-negative breast cancer (TNBC), and may guide potential therapeutic strategies.

## 5. Conclusions

Our research reveals the critical role of Mycn in pubertal mammary gland development, particularly in promoting the activation of quiescent mammary stem cells maintained by Bcl11b. Through single-cell RNA sequencing and functional experiments, we found that Mycn is spatially enriched in terminal end buds (TEBs), especially in basal cells, where it promotes cell proliferation and regulates ductal morphogenesis. At the mechanistic level, Mycn influences stem cell maintenance and differentiation by regulating the expression of key downstream target genes, including Bcl11b and Tspan8. The deletion of Bcl11b leads to significant downregulation of Tspan8 expression, further illustrating the functional interaction between these two regulatory factors in stem cell activation and lineage commitment.

More importantly, our research emphasizes that Mycn expression levels must be precisely regulated to maintain homeostasis during mammary gland development. We found that Mycn deficiency or low expression impairs the regenerative capacity of mammary stem cells, leading to underdeveloped mammary trees with reduced branching, affecting normal mammary structure and function. Conversely, Mycn overexpression, while promoting cell proliferation, simultaneously suppresses the expression of quiescent stem cell markers and regulators (such as Bcl11b and Tspan8), thus disrupting stem cell homeostasis, resulting in decreased regenerative capacity, and potentially inducing abnormal proliferation and tumorigenesis.

Given Mycn’s oncogenic potential in various cancers, including triple-negative breast cancer, our research further suggests that Mycn not only plays a core regulatory role in mammary development, but that its expression imbalance may also be an important driving factor in cancer occurrence. This provides new insights into understanding the molecular mechanisms of mammary development and the initiation pathways of breast cancer.

However, our research has certain limitations. The complex interaction network among Mycn, Bcl11b, and Tspan8 requires further investigation, especially regarding the regulatory mechanisms of direct binding sites or mediating factors. Additionally, although we observed Mycn-regulated phenotypes in mouse models, its translational relevance in human mammary development and disease needs further verification.

Future research should focus on analyzing the regulatory network among Mycn, Bcl11b, and Tspan8 at the transcriptional and epigenetic levels. Exploring whether pathways regulated by Mycn in mammary development are abnormally activated in breast cancer, as well as the expression correlation of Mycn, Bcl11b, and Tspan8 in human breast cancer samples, will be of significant value. Furthermore, using lineage tracing models can determine whether changes in Mycn expression levels affect the long-term fate of stem cells, potentially leading to stem cell depletion or malignant transformation. Finally, whether regulating Tspan8 or Bcl11b could serve as potential strategies for intervening in Mycn-driven abnormal proliferation or tumorigenesis is also worth further validation in animal models.

Ultimately, our research establishes the key regulatory role of Mycn in the delicate balance between quiescence, activation, and differentiation of pubertal mammary stem cells. Precise regulation of Mycn is crucial for normal mammary development, and its dysregulation may promote the occurrence of breast cancer. These findings open new directions for understanding mammary development mechanisms and developing targeted therapeutic strategies for breast cancer.

## Figures and Tables

**Figure 1 cells-14-01239-f001:**
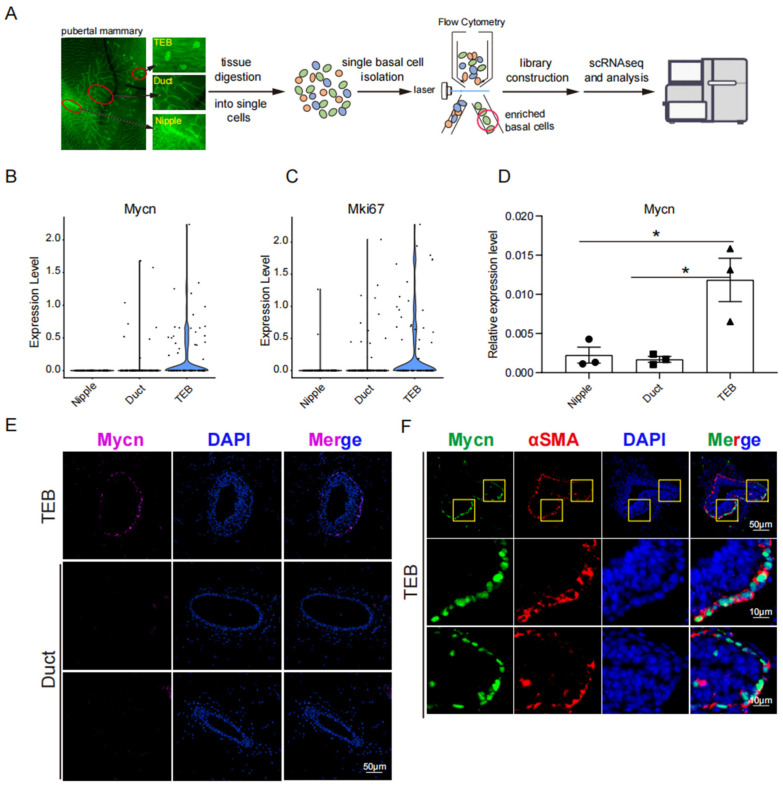
Expression of Mycn in the developing mouse mammary gland. (**A**) Schematic workflow for single-cell RNA-Seq in (**B**,**C**). Distinct mammary tissues, including nipples, ducts and TEBs, from 5-week-old *K14-Cre/Rosa26-mTmG* mice were micro-dissected and processed into single-cell suspensions for single-cell SMART-Seq. The area circled by the red circle indicates the isolated pure basal population. (**B**,**C**) Violin plots showing the expression of transcription factor Mycn (**B**) and the proliferation marker Mki67 (**C**) in basal cells in TEB, duct, and nipple. (**D**) Real-time PCR analysis showing the relative mRNA level of Mycn in TEB, duct, and nipple basal cells normalized to the expression of β-actin. Statistical analysis was performed using a two-tailed unpaired *t*-test. Data is presented as mean ± SEM, *n* = 3. * *p* < 0.05. Each circle, square, or triangle in this bar chart represents a sample from the corresponding group. (**E**,**F**) Representative immunofluorescence imaging of the expression of Mycn in duct and TEB. Magenta/GFP: Mycn; Red: αSMA; blue: DAPI. And the portion highlighted by the yellow box in the upper channel of the subfigure (**F**) is magnified in the lower channel. Scale bar, 10 μm or 50 μm.

**Figure 2 cells-14-01239-f002:**
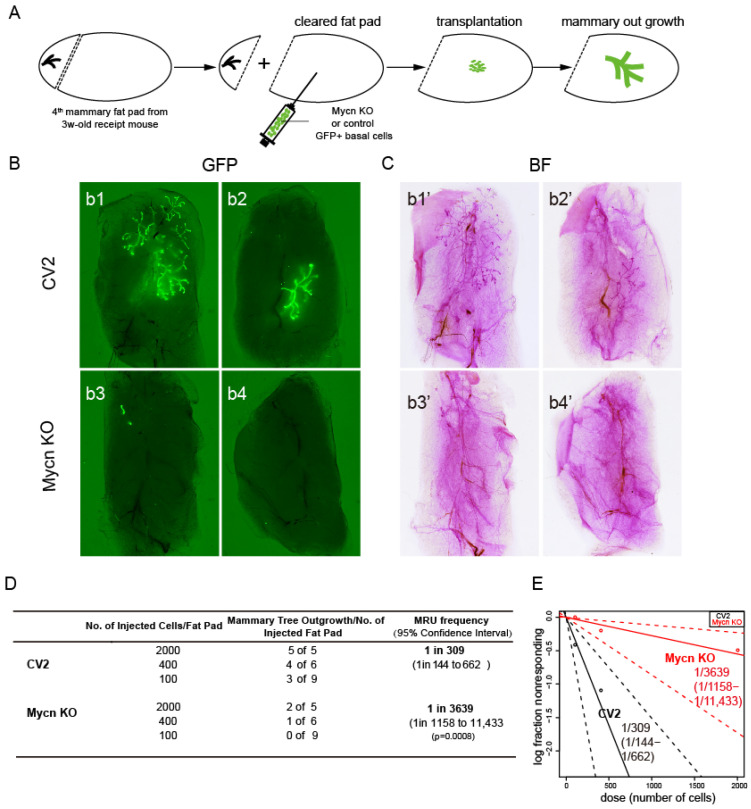
*Mycn*-deficient basal cells were defective in mammary gland reconstitution activity. (**A**) Schematic diagram illustrating the process of mammary cell transplantation. The *Mycn* gene was knocked out (KO) using CRISPR in basal cells isolated from the mammary glands of 6-week-old C57BL/6 wild-type (WT) female mice. Subsequently, the *Mycn* KO and control (CV2) GFP-positive basal cells were injected into the cleared 4th mammary fat pads of 3-week-old recipient C57BL/6 WT female mice and collected 6–8 weeks after transplantation. (**B**) Representative immunofluorescence images under a stereomicroscope showing the mammary outgrowths of both the *Mycn* KO and control groups after transplantation. (**C**) Wholemount carmine stain images showing the mammary outgrowths in (**B**). (**D**) Data table showing the numbers of total injections, outgrowths, and repopulating frequency of *Mycn* KO and control basal populations in the cleared fat pad transplantation assay. (**E**) ELDA plot of limiting dilution transplant assay showing the frequency of the repopulating unit in the two different subpopulations of *Mycn* KO and control basal cells.

**Figure 3 cells-14-01239-f003:**
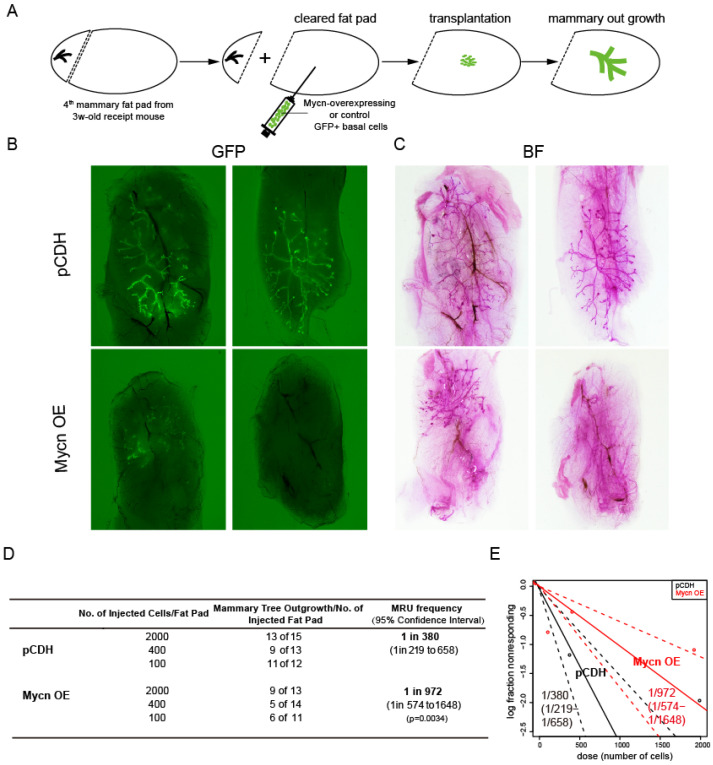
*Mycn*-overexpressing basal cells also displayed diminished mammary gland regeneration capacity. (**A**) Schematic diagram illustrating the transplantation process of of *Mycn*-overexpressing and control (pCDH) basal cells. These two groups of basal cells were injected into the cleared 4th mammary fat pads of 3-week-old recipient C57BL/6 WT female mice and analyzed 6–8 weeks after transplantation. (**B**) Representative immunofluorescence images viewed under a stereomicroscope showing the mammary outgrowths in both the *Mycn* overexpressing and control groups following transplantation. (**C**) Whole-mount carmine stain images visualizing the mammary outgrowths shown in (**B**). (**D**) Data table presenting injection numbers, outgrowths counts, and repopulating frequency of *Mycn*-overexpressing and control basal populations in the cleared fat pad transplantation assay. (**E**) ELDA plot of the limiting dilution transplant assay showing the frequency of the repopulating units within the distinct subpopulations of *Mycn*-overexpressing and control basal cells.

**Figure 4 cells-14-01239-f004:**
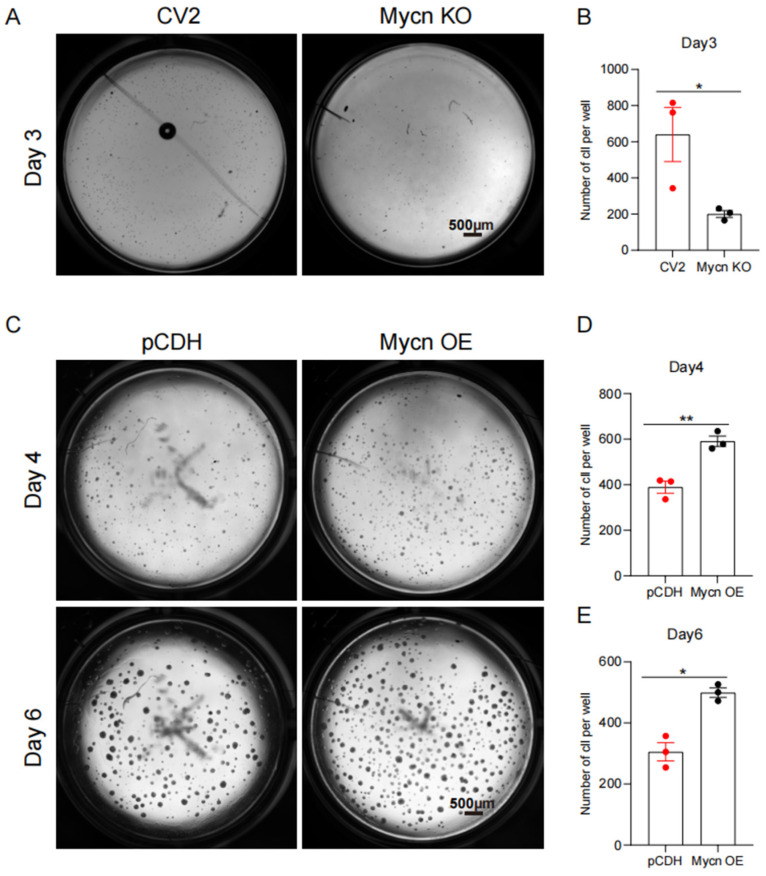
*Mycn* deficiency inhibited, while overexpression promoted, basal cell proliferation. (**A**) Colony formation assay for *Mycn* KO and CV2 basal cells. Five thousand cells per well were seeded for each group and cultured for 3 days. Scale bar, 500 μm. (**B**) Bar chart showing the average cell numbers for the cultured cells on Day3, as shown in (**A**). (**C**) Colony formation assay for *Mycn*-overexpressing and control (pCDH) basal cells. Five thousand cells per well were seeded for each group and cultured for 4 days and 6 days. Scale bar, 500 μm. (**D**) Bar chart showing the average cell numbers for the cultured cells on Day4, as shown in (**C**). (**E**) Bar chart showing the average cell numbers for the cultured cells on Day6, as shown in (**C**). Each red or black solid circle in these bar charts in subfigure (**B**,**D**,**E**) represents a sample from the corresponding group. All the above statistical analysis was performed using a two-tailed unpaired *t*-test. Data is presented as mean ± SEM, *n* = 3. * *p* < 0.05, ** *p* < 0.01.

**Figure 5 cells-14-01239-f005:**
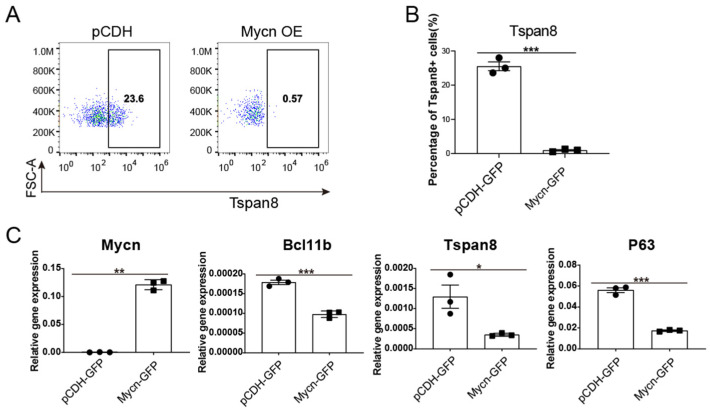
*Mycn* overexpression downregulated the quiescent stem cell regulators Bcl11b and Tspan8. (**A**) Representative FACS plots showing the reduced expression level of Tspan8 in the *Mycn*-overexpressing group compared with the control group basal cells. (**B**) Statistic analysis of the percentage of Tspan8+ basal cells in both *Mycn*-overexpressing and control (pCDH) basal populations, as detected by FACS in (**A**). (**C**) Real-time PCR analysis showing the relative mRNA level of several key stem cell regulators in *Mycn*-overexpressing and control basal cells, normalized to the expression of *β-actin*. Each circular or square marker in these bar charts in subfigure (**B**,**C**) represents an individual sample from the corresponding group. Statistical analysis was performed using a two-tailed unpaired *t* test. Data is presented as mean ± SEM, *n* = 3. * *p* < 0.05, ** *p* < 0.01, *** *p* < 0.001.

**Figure 6 cells-14-01239-f006:**
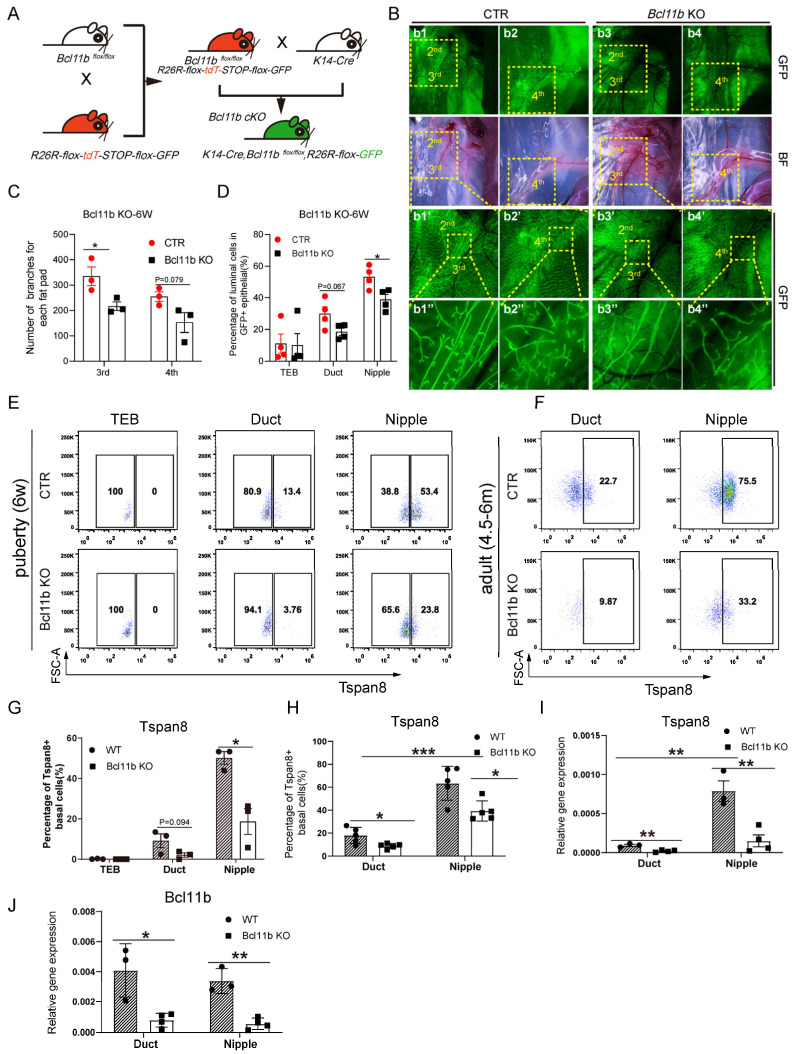
*Bcl11b* loss downregulates Tspan8 expression. (**A**) Schematic diagram illustrating the breeding strategy to generate *Krt14* promoter-specific *Bcl11b* conditional knockout (KO) mice. Through crossing *Bcl11b*^flox/flox^/*mTmG* mice with *K14-Cre* mice, offspring with the genotype K14-*Cre*/*Bcl11b*^flox/+^/*mTmG* were obtained. Subsequent backcrossing of this progeny with *Bcl11b*^flox/flox^/*mTmG* mice generated *K14-Cre*/*Bcl11b*^flox/flox^/*mTmG* mice, enabling specific ablation of *Bcl11b* in *Krt14*-positive basal epithelial cells of the murine mammary gland. (**B**) Representative immunofluorescence and bright field images of the 2nd, 3rd, and 4th mammary fat pads demonstrating reduced mammary branches after *Bcl11b* KO in 6-week-old female C57BL/6 mice compared with the control mice. (**C**) Bar chart depicting the effect of conditional *Bcl11b* KO on mammary branches of different mammary fat pads compared with the control group. Statistical analysis was performed using a two-tailed unpaired *t*-test. Data are presented as mean ± SEM, *n* = 3. * *p* < 0.05. (**D**) Bar chart illustrating the effect of conditional *Bcl11b* KO on mammary epithelial composition compared with the control group. Statistical analysis was performed using a two-tailed unpaired *t*-test. Data are presented as mean ± SEM, *n* = 3. * *p* < 0.05. (**E**,**F**) Representative FACS plots demonstrating the expression of *Bcl11b* KO on the expression of Tspan8 in the TEB, duct, and nipple basal cells of 6-week-old pubertal mice (**E**) or in the duct and nipple basal cells of 4.5–6-month-old adult mice (**F**) compared with the control mice. All the cells analyzed in FACS were GFP-positive basal epithelial cells. (**G**) Statistic analysis of the FACS of the percentage of GFP-labeled Tspan8+ basal cells in 6-week-old *Bcl11b* KO and control mice, as shown in (**E**). (**H**) Statistical analysis of the FACS of the percentage of GFP-labeled Tspan8+ basal cells in 4.5–6-month-old *Bcl11b* KO and control mice, as shown in (**F**). (**I**) Real-time PCR analysis showing the effect of *Bcl11b* KO on the relative mRNA level of *Tspan8* in duct and nipple basal cells normalized to the expression of *β-actin* in adult 4.5–6-month-old mice compared with the control mice. Statistical analysis was performed using a two-tailed unpaired *t*-test. Data are presented as Mean ± SEM, *n* = 3–4. * *p* < 0.05, ** *p* < 0.01. (**J**) Real-time PCR analysis showing the relative mRNA level of *Bcl11b* in duct and nipple basal cells, normalized to the expression of *β-actin* in adult 4.5–6-month-old mice compared with the control mice. Statistical analysis was performed using a two-tailed unpaired *t*-test. Data is presented as mean ± SEM, *n* = 3–4. * *p* < 0.05, ** *p* < 0.01, *** *p* < 0.001.

**Figure 7 cells-14-01239-f007:**
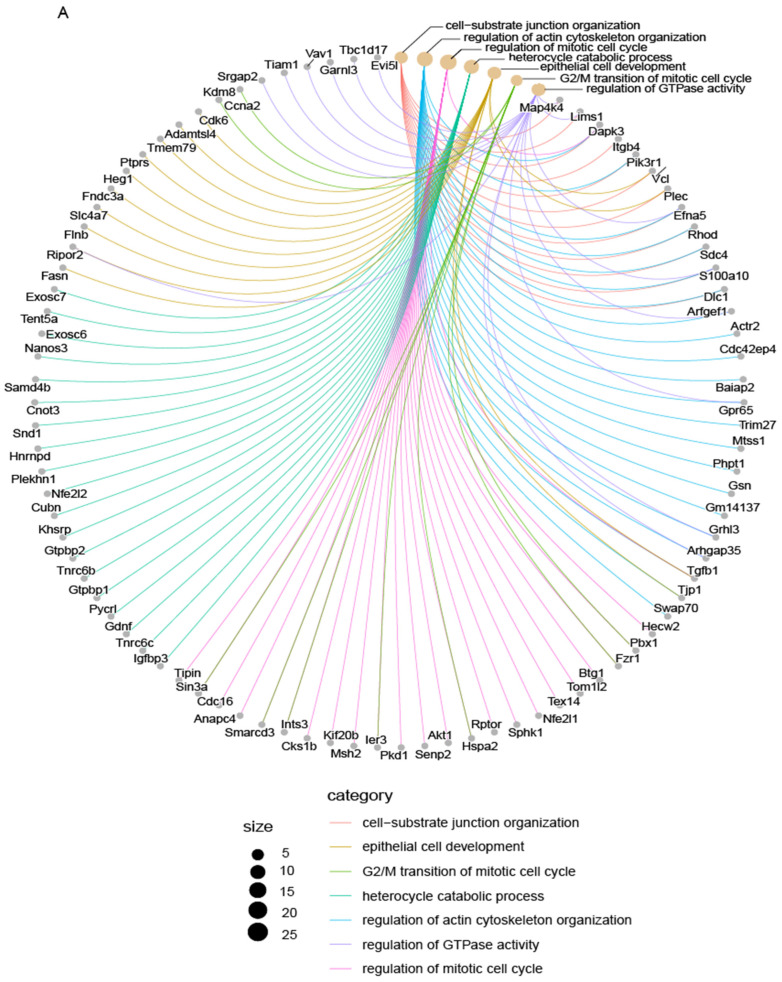
Mycn regulated cell proliferation and differentiation simultaneously. Detection of Mycn binding sites using the D-CUT&RUN method in Comma Dβ cells. (**A**) Cnetplot showing KEGG pathway analysis of Mycn target genes in 2 × 10^6^ Comma Dβ cells. (**B**) Emapplot showing biological processes from GO pathway analysis of Mycn target genes in Comma Dβ cells. (**C**) Biological processes fromGO pathway analysis of Mycn target genes in Comma Dβ cells.

**Figure 8 cells-14-01239-f008:**
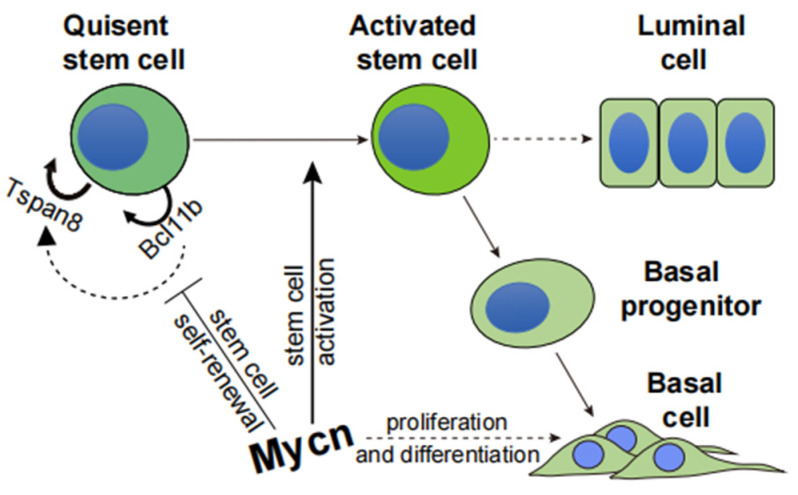
The model for Mycn’s role in the activation of quiescent mammary stem cells. Mycn, a transcription factor and regulator of cell proliferation, serves a fundamental role in mammary gland development and regeneration. Mycn exhibits high expression in TEB basal cells, which are actively engaged in cell proliferation and differentiation. *Mycn* KO and *Mycn*-overexpressing basal epithelial cells demonstrated that Mycn enhances cell proliferation and differentiation, indicating that optimal Mycn expression levels are essential for mammary gland regeneration. Additionally, Mycn facilitates quiescent stem cell activation through downregulation of key quiescent stem cell markers and regulators, particularly Bcl11b and Tspan8. The ChIP seq data validated that Mycn coordinates cell proliferation and differentiation. In vivo, the specific knockout of *Bcl11b* in basal epithelial cells markedly reduced Tspan8 expression and enhanced cell proliferation. The research identified a significant signaling axis, Mycn/Bcl11b/Tspan8, which regulates the activation and differentiation of quiescent stem cells in the mammary gland. Thus, Mycn functions as a critical regulator in mammary gland development and regeneration, activating quiescent stem cells and promoting cell proliferation and differentiation through Bcl11b and Tspan8 downregulation. These findings present important implications for understanding and potentially treating diseases and cancers characterized by Mycn overexpression.

## Data Availability

The data that support the findings in this study are openly available in GEO: The original scRNA-seq data generated in this study can be accessed via accession number GSE251933, while published datasets used for analysis are available under accession numbers GSE151426 and GSE94782.

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
