# Peer review of "Mycn Is Essential for Pubertal Mammary Gland Development and Promotes the Activation of Bcl11b-Maintained Quiescent Stem Cells"

_cells, 2025, doi:10.3390/cells14161239_

Round 1

Reviewer 1 Report

Comments and Suggestions for Authors

Overall, this is a neat piece of work that digs into how Mycn helps kick quiescent mammary stem cells into gear during puberty. You use single‑cell RNA‑seq to show Mycn lighting up in those terminal end buds, and then CRISPR knock‑out and overexpression coupled with transplantation and colony assays to tease out how too little or too much Mycn throws stem cell function off balance. You tie that back to known quiescence regulators Bcl11b and Tspan8, and even do some CUT&RUN to show Mycn’s genome‑wide footprints. The story really comes together around the idea that you need just the right dose of Mycn to get ductal outgrowth without burning through your stem cell pool.

I did spot a couple of slips you might want to clean up. In the introduction you refer to “Tripple‑negative breast cancer” instead of “triple‑negative,” and throughout there are places where citations mash up with words, for example “mammary1,11” should have a space or superscript formatting, and similarly elsewhere the spacing around citations gets lost. Also in Figure legends the term “Mycn‑Deficient” is sometimes hyphenated and elsewhere not, so maybe pick one style.

This is a cancer study, I recommend opening your report with a cohesive paragraph that introduces cancer biology and treatment modalities, cite review by Chief of US National Cancer Institute D. Sonkin, A. Thomas “Cancer Treatments: Past, Present, and Future (2024)” . It wouldn’t hurt to beef up the intro a bit by spelling out why quiescence matters beyond stem‑cell longevity, maybe a sentence on how misregulated exit from quiescence can drive hyperplasia or tumorigenesis. And in the discussion you summarize nicely but you could expand on limitations and next steps. You must discuss the pro and con of single cell analysis. Bulk sequencing remains cost-effective and robust for detecting average gene-expression shifts across large cohorts but can mask intra-tumoral heterogeneity and rare cell states. In contrast, single-cell methods unveil cellular diversity and uncover minority populations, yet they often suffer from increased technical noise, higher per-cell costs, and complex analytic biases. Recent reviews such as “Technical and Biological Biases in Bulk Transcriptomic Data Mining for Cancer Research” (2025) highlight how sample composition and batch effects can skew bulk analyses, while “Genetic Expression in Cancer Research: Challenges and Complexity” (2024) emphasizes drop-out events and amplification biases inherent to bulk and single-cell protocols. By citing and discussing these perspectives, our study clarifies why we chose a hybrid strategy, using bulk profiles for initial subtype discovery and single-cell validation to ensure that our radiomic,genomic correlations truly reflect the underlying cellular architecture. For example, what happens if you tweak Mycn levels in adult glands or during pregnancy, and how might this translate to human TNBC? Another paper of breast cancer single cell analysis should be mention and iscuss is “Identification of the novel exhausted T cell CD8 + markers in breast cancer, 2024”. A short paragraph on potential off‑target effects of CRISPR or the need for in vivo lineage tracing would round things out. A little more back‑and‑forth with the broader Myc literature, in particular, how your findings compare with what’s known about c‑Myc in normal mammary tissue, would also help anchor your work in the field. Otherwise, it’s a solid study that I enjoyed reading.

Reviewer 2 Report

Comments and Suggestions for Authors

In this study, Lin Z et al used single cell RNA profiling and force biology including CRISPR knockdown and overexpression to study the role of Mycn gene in mammary gland development. They found that balancing appropriate levels of Mycn expression is crucial for normal development and preventing tumorigenesis. Since deregulated expression of Mycn oncoprotein has been found in many cancers, this study provided important insights into the role of Mycn in normal development and added additional baseline knowledge for cancer research. See a few minor comments/suggestions in below:

  • Why the Supplementary Fig. 1A was mentioned in the Introduction section instead in the beginning of the results section?
  • In Figure 4, Mycn knockdown showed deceased colony forming. Do you have data/figure (PCR or western blots) to show the efficacy of the Mycn knockdown?
  • In figure 5A, the text labelling for both Y and X axis are too small to read.
  • The rational and importance for investigating whether Bel11b can regulate Tspan8 is not clear in the manuscript. Is Bel11b a transcriptional regulator and binds to Tspan8 promoter for regulating its gene expression?  

Round 2

Reviewer 1 Report

Comments and Suggestions for Authors

thepaper is been improved